# Visitors' Willingness to Pay for Protected Areas: A New Conservation Donation in Aso Kuju National Park

Thomas Edward Jones [1], Duo Xu [2], Takayuki Kubo [1] and Minh-Hoang Nguyen [3,*]

1    Graduate School of Asia Pacific Studies, Ritsumeikan Asia Pacific University, Beppu 874-8577, Japan; takkubo@apu.ac.jp (T.K.)
2    London School of Economics and Political Science, London WC2A 2AE, UK; xuduo0201@outlook.com
3    Centre for Interdisciplinary Social Research, Phenikaa University, Yen Nghia Ward, Ha Dong District, Hanoi 100803, Vietnam
*    Correspondence: hoang.nguyenminh@phenikaa-uni.edu.vn

**Abstract:** Protected areas (PAs) such as national parks face funding issues that undermine effective management. Therefore, many PAs are exploring new financial instruments, such as visitor donations, to supplement their conservation budgets. This paper investigates visitor perceptions of one such system, a new conservation donation under consideration in Aso Kuju National Park, southwest Japan, is due to be introduced. Our on-site survey at two trailheads in autumn 2022 gauged visitors' willingness to pay (WTP) the expected JPY 500 donation. The analysis used Bayesian linear regression to look for significant predictors of WTP. Findings show that female, older, and higher-income visitors were more likely to pay the donation collectively rather than voluntarily. Prior knowledge of the donation system was also a significant predictor of WTP, but more frequent climbers were significantly less likely to pay the donation collectively, regardless of prior Kuju climbing experience, possibly due to the perceived increase in use costs. Moreover, visitors willing to pay the cooperation donation collectively are also willing to pay higher prices than those willing to pay voluntarily. The elicited WTP values confirm that the implementation of a new conservation donation could help to improve the long-term sustainable financing of PAs such as Aso Kuju while raising issues over price fairness.

**Keywords:** willingness to pay; nature-based tourism; Aso Kuju National Park; donation; Japan

## 1. Introduction

In recent years, demand for nature-based tourism has increased, with a strong recovery after the initial hiatus of the COVID-19 pandemic [1,2]. There has been a reciprocal increase in the territory designated for conservation within extended "protected area" (PA) networks, but many PA destinations still suffer from "over-tourism" via negative impacts, including pollution of air and water, littering, or congestion [3]. PAs must manage such tourism pressure and seek sustainable solutions via economic or regulatory instruments that include direct fees (such as entrance tickets) or indirect charges for car parks or other facilities, including local or national taxes (e.g., value-added tax) [4]. However, Japan's PAs do not charge an entrance fee per se but are exploring new financial instruments, such as visitor donations, to supplement their conservation budgets [5]. Examples exist in mountainous PAs, such as Mt. Fuji (since 2013) or Yakushima (since 2017). Prior to the expected establishment of a new conservation donation in Aso Kuju National Park, southwest Japan, our paper investigated visitor perceptions and willingness to pay (WTP) the expected JPY 500 donation. An on-site survey at two trailheads in autumn 2022 gauged visitors' stated preferences, while the analysis used Bayesian linear regression to seek out significant predictors of WTP. Sections 2–4 of the paper provide background information regarding the related literature, the study site, and the methodology used for data analysis. The fifth section presents the results, followed by the discussion and conclusions for PA management.

## 2. Literature Review

### 2.1. Economic Valuation for the Environment

Economic valuation is the process of assigning specific monetary values to non-market assets, goods, and services that positively contribute to people's well-being. Individuals' willingness to pay (WTP) for such goods or services determines their contribution to people's well-being [6]. Economic valuation of the environment can measure individuals' WTP for resources or changes to them, and the objective is to determine the total economic value (TEV) associated with the relevant commodity. Generally, the TEV of the environment consists of use value (UV) and non-use value (NUV), wherein the distinction lies in the directness of the individual–environment interaction [7]. Use values pertain to the utilisation and appreciation of nature through both extractive and non-extractive encounters (direct use value), appreciation of ecological functions (indirect use value), and the prospective utilisation of nature in the future (option values). Conversely, non-use values encompass the contentment individuals derive from the mere existence of nature (existence value) and from the knowledge that others can derive benefits from nature (bequest value) [8].

Various methods have been developed by economists to capture the TEV of environmental resources, with direct and indirect approaches being widely utilised. According to Laurila-Pant, et al. [9], direct methods employ survey and experimental techniques to elicit preferences, often utilising questionnaires in which respondents indicate their willingness to pay to enjoy and/or conserve resources. Indirect methods use observed market-based information, such as Hedonic Price techniques, to infer preferences. Hedonic pricing is one of the commonly employed methods, along with the contingent valuation method (CVM) [10]. CVM is suitable for estimating both use- and non-use-values, capturing people's preferences for public goods by ascertaining individuals' WTP for specific improvements [11,12]. Typically, CVM involves data collection through mail surveys, questionnaires, or interviews, where interviewees are required to express their WTP for a certain non-market good or service. It has been adopted in various areas, such as studies of city greening tourism, endangered species, ecological system service, and environmental policy establishment [13]. Conversely, hedonic pricing methods focus on analysing costs and prices, primarily used to estimate the demand for goods [14] or analysis of the housing market [15], green spaces [16], and amenity values of urban ecosystems [17].

### 2.2. WTP for Protected Areas: The Global Context

Protected areas are expected to play a key role in curbing the significant degradation of natural resources and biodiversity. Since the establishment of the world's first national park in 1872, the terrestrial and inland waters protected area and other effective area-based conservation measures (OECMs) coverage has increased to 17.28% [18]. However, for effective management of expanding protected areas, sustainable finance is required. One of the potential financial sources for protected area management and conservation is directly from visitors. The estimated annual count of tourist visits to terrestrial protected areas stands at 8 billion, resulting in approximately USD 600 billion in direct in-country spending each year, according to Balmford, et al. [19] Additionally, Buckley, et al. [20] conducted an assessment of the economic worth of protected areas for the enhanced mental well-being of visitors, yielding a conservative global estimate of USD 6 trillion annually.

However, visitors' valuation of nature is different, so their willingness to pay for use and non-use values in protected areas is also dissimilar [21]. Willingness to pay (WTP) analysis is, therefore, widely employed in the study of protected areas such as national parks, offering insights to develop effective pricing policies and marketing strategies. For instance, Bhat and Sofi [10] gauged households' WTP for biodiversity conservation in India's Dachigam National Park. Their findings revealed respondents' WTP for park improvements, with a mean value of INR 245.57 (USD 3.32) per year. Likewise, Khan [22] investigated the visitors' WTP for visiting Margalla Hills National Park near Islamabad. The study highlighted that enhancing the park's quality could result in a 39% increase in leisure benefits. It also revealed that an INR 20 per person park entrance fee could

maximise total revenue (ibid). With a focus on nature conservation, White and Lovett [23] employed a contingent valuation interview to examine visitors' WTP for an annual amount in taxes for national parks in the UK. The results indicated that a significant portion of park visitors would be willing to make additional donations for nature conservation. Furthermore, a study by Song, et al. [24] evaluated visitors' WTP entrance fees based on a dataset comprising 1215 individuals visiting China's planned Qinling National Park. The investigation showed that predictors such as gender, education level, income, and frequency of trips to natural attractions strongly influenced visitors' WTP, along with concerns about commercial exploitation [24]. These studies reflect the global scope of WTP analysis as a research domain and the importance of identifying significant predictors that can capture the demand dynamics across diverse visitor profiles.

### 2.3. WTP in Japanese National Parks

Japan's "multi-purpose" nature parks have no entrance fee system per se, but require visitors to pay for certain in situ services, such as parking or camping etc. [25]. This de facto reliance on public funding to cover management and operation costs has been strained as maintenance issues have increased [5]. Implementing a "voluntary fee" could reduce the financial burden on government coffers and enhance environmental conservation. Mt. Fuji pioneered this approach with a donation of JPY 1000 introduced in 2013 aimed at improving climber safety and environmental conservation [26]. While the donations are primarily requested from climbers ascending above the 6th station, they are collected from anyone who wishes to contribute to Fuji's overall conservation. The funds are utilised to implement and enhance activities and services related to environmental conservation, climber safety, and the provision of information. In 2017, Shizuoka Prefecture received 54,087 donations totalling JPY 52,047,583 (USD 377,983), while Yamanashi Prefecture received 98,254 donations amounting to JPY 96,704,776 (USD 701,950) [27]. Parallel systems have been introduced at other mountainous parks, such as Yakushima, and a JPY 500 donation is under consideration at Aso Kuju National Park (personal correspondence with a park ranger).

## 3. A New Conservation Donation in Aso Kuju National Park

Japan has 34 national parks, and Aso Kuju, designated in 1934, is among the oldest of them. Located in the centre of Kyushu, Aso Kuju spans Kumamoto and Oita prefectures (Figure 1). The park covers 727 square kilometres, including Mt. Aso, one of Japan's largest—and most active—volcanoes, and the Kuju range, which includes the highest peak on the island of Kyushu. A network of hiking trails with spectacular views and natural features enables visitors to enjoy volcanic landscapes and characteristic rural scenery that was recognised by UNESCO as a World Agricultural Heritage Site in 2013 and a Global Geopark in 2014 [28].

### 3.1. Natural Environment of Aso Kuju National Park

The Aso caldera encompasses notable topography, such as the Mt. Nakadaka crater, the Komezuka volcanic cone, and surrounding Kusasenri-ga-hama grasslands. Meanwhile, the Kuju mountain range includes such volcanic landscapes as the meadows, moors, and wetlands of Tadewara and Bogatsuru. Mt. Tsurumi and Mt. Yufu delineate the park's northeastern limits with peaks offering panoramic vistas of the Beppu Bay and the Yufuin Basin. The park's ecology can be broadly classified into three altitudinal zones: solfataric wilderness characterises the upland areas, interspersed with grasslands and forests that dominate the foothills. The solfataric upland zone hosts biota such as Kyushu azalea (*Rhododendron kiusianum*) and Cowberry (*Vaccinium vitis-idaea*) that have adapted to cope with the effect of volcanic gases. Less harsh environments at lower altitudes encourage greater biodiversity in grassland areas where traditional land use practices such as pasturing, controlled burning, and mowing are employed to prevent forest succession. Pampas grass prevails in these moors and mountain meadows that form habitats for

rare plants and butterfly species such as *Shijimiaeoides divinus* [29]. Forests host endemic sub-species, such as the Kyushu-ezozemi cicada, while the grasslands support the Daikoku-kogane (a variety of scarab beetle) and bird species, such as Meadow Bunting (*Emberiza cioides*) and Black-Browed Reed Warbler (*Acrocephalus bistrigiceps*).

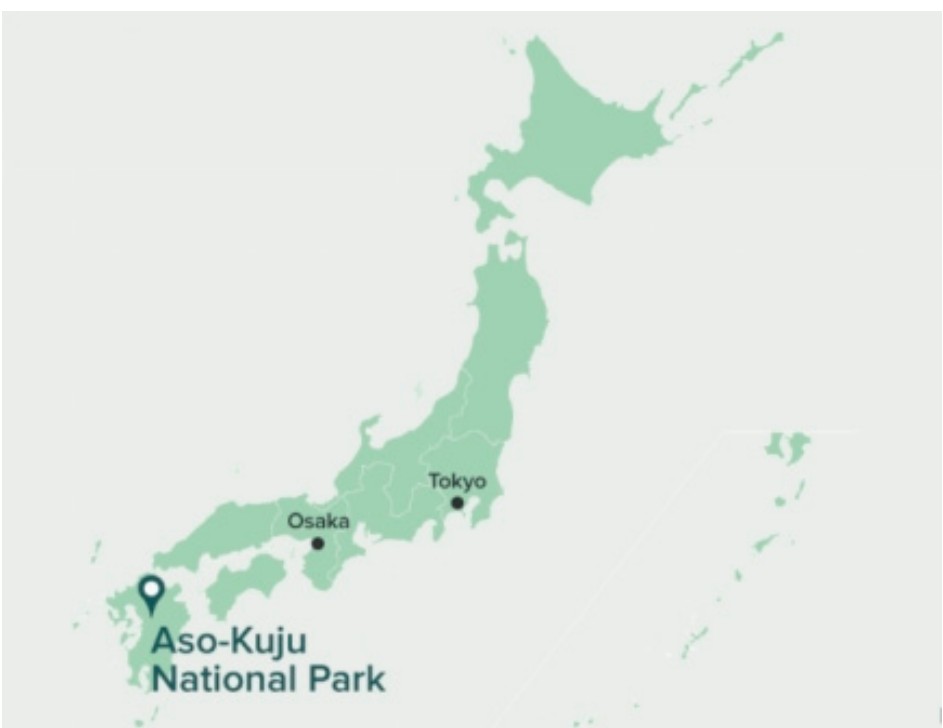

**Figure 1.** The map of Aso Kuju National Park [30].

### 3.2. Visitor Access and Activities

Located between Kumamoto and Oita prefectures in central Kyushu, Aso Kuju National Park attracted over 17 million visits in 2018 [31]. Public transport includes buses and trains, but most visitors use private or rental cars to explore the Kyushu Odan highway that traverses the park. Common nature-based tourism activities are driving or hiking through volcanic landscapes from mountain meadows and calderas to forests, waterfalls, and wetlands [32]. The abundance of volcanoes is accompanied by hot springs, which contribute to the allure of many prestigious *onsen* towns in and around the park. Visitors can also participate in seasonal festivals and cultural events, such as the controlled burn (*noyaki*), consisting of traditional rituals to burn off the stubble and prevent the succession of grassland ecosystems into the forest.

### 3.3. Profile of Visitors to the Kuju Mountain Range

Kuju's picturesque landscapes, volcanic peaks, grasslands, and mountain marshes attract a diverse range of visitors dominated by domestic and interregional trips. Typical tourist 'pull-factors' include sightseeing, family trips, mountain climbing, and soaking in hot springs. Kyushu is a popular destination for Japanese tourists across various age groups, ranging from teenagers to octogenarians, but domestic park visits are dominated by an older profile [32]. According to a 2019 survey, 85% of them were repeat visitors to the national park. There is also an increasing presence of foreign tourists, most of whom are visiting the park for the first time. Other differences included a prevalence of Japanese car visitors, whereas international visitors were more likely to use public transport. Significant differences in variables such as the highest level of completed educational degree could predict visitors' WTP for the new conservation donation, although "enjoying nature

with family or friends" was a common motivation for both domestic and international segments [32].

### 3.4. Environmental Impacts and Conservation Donation in Kuju

At Kuju, the number of overnight stays did decline during the pandemic, but the daily footfall during peak periods has increased at certain "honeypot" destinations. As visitors crowded into particular spaces during certain seasons, environmental impacts have occurred due to a growing number of day trippers. So-called 'overuse' issues include excessive pressure on specific facilities such as car parks and toilets or disruption of the ecosystem due to air and noise pollution from automobile use. In addition, the mountain meadows require considerable financial input to maintain the 'secondary nature' grassland through annual events to burn off the stubble. Like other mountainous PAs, park rangers at Kuju are exploring a new visitor donation of JPY 500 to supplement their conservation budgets. Two donation options are under consideration: social obligation (i.e., all climbers make financial contributions collectively) and optional donation (i.e., only individuals who want to pay). Thus, the study aims to examine the predictors of the donation types, providing insights that can support donation policymaking and implementation.

## 4. Materials and Methods

### 4.1. Data Collection and Sample

Our study site was the Kuju region of the national park, on Oita Prefecture's western border. Surveys were collected over four days from October–November 2022 timed to coincide with the autumn leaves' peak season when the vibrant fall foliage and clement weather conditions attract a visitation plateau. Primary data was collected through an on-site, self-administered questionnaire survey led by two professors (authors TEJ and TK), together with five trained data collectors. A pre-survey pilot test was conducted on university students and questions were back-translated from English into Japanese. Two checkpoints were subsequently set up on the most popular gateway trails around the Chojabaru Visitor Centre (approx. 1000 m ASL) and Makinoto Pass (approx. 1300 m ASL). The latter is a mountain pass and car park used by most hikers climbing up to the Kuju peaks, while Chojabaru is a tourist hub and visitor centre adjacent to the Tadewara highland marsh that attracts the most visitors to Kuju overall. By selecting these two trailheads as study sites, our 606 valid responses could claim saturation by representatively covering hiker, climber, and sightseer segments. Respondents filled in the questionnaire independently, but survey staff were present throughout to answer queries and ensure all the questions were completed. Before administering the survey, a brief explanation of the questionnaire's purpose and scope was provided to potential participants, who also had the right to refuse their responses.

### 4.2. Data Treatment and Analysis

Bayesian linear regression analysis was employed in the current study for several reasons. First, Bayesian inference treats all the quantities probabilistically, including the known and unknown quantities. Any unobserved data or unknown parameters can be considered unknown quantities [33]. Thus, Bayesian inference enables researchers to construct parsimonious models and focus solely on the issues of interest [34]. Second, the Bayesian analysis aided by Markov Chain Monte Carlo (MCMC) algorithms allows computing the posterior distributions of complex models, such as hierarchical and non-linear models [35,36]. In the current study, Model 2 contains noncollinearity, so the great model-fitting capability of Bayesian analysis aided by MCMC algorithms is required. Third, Bayesian analysis's estimation and visualisation of credible intervals are a good alternative for the dichotomous *p*-value approach, which is suggested to be a crucial reason behind the reproducibility crisis [37,38]. In particular, Bayesian analysis generates credible intervals in which the estimated parameters are random and bounds are fixed. The credible

intervals help demonstrate the region where the true parameter value has high probability to fall within.

We followed these four steps to perform Bayesian analysis [39]: (1) model construction, (2) model fitting, (3) model diagnoses, and (4) result interpretation. The analysis was conducted using the bayesvl R package, which offers researchers a user-friendly and intuitive protocol, the ability to visualise graphics, and cost-effectiveness [40].

In the first step, we constructed Model 1 to examine whether visitors' willingness to pay the collective cooperation donation affects the amount that they are willing to pay. The asterisk ('*') in equation is arithmetic operator meaning multiplication.

$$Donation\,Amount \sim normal(\mu, \sigma) \tag{1}$$

$$\mu_i = \beta_0 + \beta_{CollectiveDonation} * CollectiveDonation_i \tag{2}$$

$$\beta \sim normal(M, S) \tag{3}$$

The probability around $\mu$ is determined by the form of normal distribution, with the standard deviation $\sigma$. The donation amount that visitor $i$ is willing to pay is indicated by $\mu_i$. $CollectiveDonation_i$ indicates whether visitor $i$ is willing to donate collectively or voluntarily [41]. Specifically, collective donation refers to the payment as a social obligation, while voluntary donation refers to the payment as an optional donation. The model has an intercept $\beta_0$ and coefficient $\beta_{CollectiveFee}$. The probability around $\beta$ is also in the form of normal distribution.

The second model was constructed to explore the factors affecting the visitors' willingness to pay the donation. To examine whether the visitors' willingness to pay varies according to their socio-demographic features, type of travel, and frequency of usage, we added variables *Sex*, *Age*, *Income*, *Stay*, and *ClimbFrequency* into the model. Insights obtained from these predictor variables were expected to improve management effectiveness. Then, given that a previous study at Mount Fuji showed a significantly greater willingness to pay among visitors with prior awareness about the donation system [42], we employed the *Awareness* variable in the model to check if the effect of prior awareness still holds in the case of Aso Kuju National Park. In addition to that, the familiarity with a place (which can be measured through the climbing experience of the visitors in Kuju) can affect people's perceptions [43], which might include their willingness to pay, so the variable *KujuClimbExperience* was also added to the model. However, *KujuClimbExperience* and *ClimbFrequency* might have a high correlation, which can lead to multicollinearity. Therefore, *KujuClimbExperience* was added to the model as a moderator of *ClimbFrequency* and *CollectiveDonation* to alleviate the risk of multicollinearity [44,45].

Model 2 comprises eight variables, which are displayed below:

$$CollectiveDonation \sim normal(\mu, \sigma) \tag{4}$$

$$\mu_i = \beta_0 + \beta_{Awareness} * Awareness_i + \beta_{Sex} * Sex + \beta_{Age} * Age_i + \beta_{Income} * Income + \beta_{Stay} * Stay_i + \beta_{ClimbFrequency} \\ * ClimbFrequency_i + \beta_{ClimbFrequency*KujuClimbExperience} * ClimbFrequency_i * KujuClimbExperience_i \tag{5}$$

$$\beta \sim normal(M, S) \tag{6}$$

$Awareness_i$ indicates whether visitor $i$ knew of the cooperation donation system; $Sex_i$ indicates visitor $i$'s biological sex; $Age_i$ indicates visitor $i$'s age; $Income_i$ indicates visitor $i$'s annual income; $Stay_i$ indicates whether visitor $i$ stayed overnight; $ClimbFrequency_i$ indicates visitor $i$'s climbing frequency; $KujuClimbExperience_i$ indicates visitor $i$'s climbing experience in Kuju. $\beta_{ClimbFrequency*KujuClimbExperience}$ indicates the coefficient of the non-additive effect of *ClimbFrequency* and *KujuClimbExperience* on *CollectiveDonation*. To avoid any multicollinearity between *ClimbFrequency* and *KujuClimbExperience*, we created the interaction variable between *ClimbFrequency* and *KujuClimbExperience* but did not add the *KujuClimbExperience* variable directly into the model.

As the current study is explorative, we employed uninformative priors for estimating the coefficients' posterior distributions [46]. The default prior setting of the bayesvl R package is uninformative prior, which is a normal distribution with mean at 0 and standard deviation at 10. All the variables utilised in the model construction are described in Table 1. Notably, the number of people unwilling to pay the cooperation donation was negligible (N = 9), so they were excluded from the analysis.

After constructing the models, we fitted them using the standard MCMC setups [39]. Specifically, the number of Markov chains used for fitting was four; the number of iterations of each chain was 5000; of these, the first 2000 iterations were used for warming up the simulation.

**Table 1.** Variable description.

| Variable | Meaning | Type of Variable | Value |
|---|---|---|---|
| *DonationAmount* | The amount that the visitor is willing to donate | Numerical | NA |
| *CollectiveDonation* | Whether the respondent is willing to donate the cooperation donation collectively or voluntarily | Binary | 1 = Paying collectively<br>0 = Paying voluntarily |
| *Awareness* | Whether the respondent knew the cooperation donation system | Binary | 1 = Yes<br>0 = No |
| *Sex* | The respondent's biological sex | Binary | 1 = Yes<br>0 = No |
| *Age* | The respondent's age | Numerical | NA |
| *Income* | The respondent's annual income | Numerical | 1 = Below JPY 2 Mil<br>2 = JPY 2–5.9 Mil<br>3 = JPY 6–7.9 Mil<br>4 = More than JPY 8 Mil |
| *Stay* | Whether the respondent stayed overnight in the national park | Binary | 1 = Yes<br>0 = No |
| *ClimbFrequency* | The respondent's climbing frequency in a year | Numerical | 1 = Less than once a year<br>2 = About once a year<br>3 = 2–5 times a year<br>4 = More than 6 times a year |
| *KujuClimbExperience* | The respondent's climbing experience in Kuju | Numerical | 1 = First time<br>2 = 2–5 times<br>3 = More than 6 times |

Subsequently, the convergence of the Markov chains was diagnosed using the effective sample size (*n_eff*) and the Gelman–Rubin shrink factor (*Rhat*) [47,48]. The Markov chains can be deemed convergent if the *n_eff* values exceed the standard threshold of 1000 and Rhat values equal 1. In addition, the convergence was also visually validated using the trace plots. Finally, we proceeded with result interpretation when the models' Markov chain convergence was confirmed.

## 5. Results

### 5.1. Descriptive Analysis

Table 2 shows the socioeconomic characteristics of the sample (N = 606). Tourists completing the survey were 55.94% male and 40.59% female. Most of the respondents were within the age range of 40 to 69 years (62.21%). Tourists with an annual income of around JPY 2–5.9 million accounted for the highest proportion, with 41.25%. Tourists earning more than JPY 6 million comprised 26.4% of the sample, and 18.65% did not report their annual income level.

Regarding the climbing frequency, most of the respondents were frequent climbers who had been climbing at least two times a year (60.89%). The number of visitors climbing Kuju at least twice a year also accounted for 62.87% of the sample. When asked whether they had heard of the cooperation donation system introduced previously in mountain national parks to conserve the environment, more than half of the respondents reported

'no' (57.92%). However, only nine visitors declined to pay the cooperation donation when asked.

**Table 2.** Description of the sample.

|  | N | % |
|---|---|---|
|  | **Sample (N = 606)** | |
| Gender | | |
| Male | 339 | 55.94% |
| Female | 246 | 40.59% |
| Age | | |
| 10's | 13 | 2.15% |
| 20's | 68 | 11.22% |
| 30's | 70 | 11.55% |
| 40's | 102 | 16.83% |
| 50's | 148 | 24.42% |
| 60's | 127 | 20.96% |
| 70's | 40 | 6.60% |
| 80's | 12 | 1.98% |
| Annual income | | |
| Below JPY 2 million | 83 | 13.70% |
| JPY 2–5.9 million | 250 | 41.25% |
| JPY 6–7.9 million | 80 | 13.20% |
| More than JPY 8 million | 80 | 13.20% |
| Stay overnight | | |
| Yes | 199 | 32.84% |
| No | 316 | 52.15% |
| Yearly climbing frequency | | |
| Less than once a year | 79 | 13.04% |
| About once a year | 63 | 10.40% |
| 2–5 times a year | 169 | 27.89% |
| More than 6 times a year | 200 | 33.00% |
| Kuju climbing experience | | |
| First time | 116 | 19.14% |
| 2–5 times | 217 | 35.81% |
| More than 6 times | 164 | 27.06% |
| Cooperation donation awareness | | |
| Yes | 247 | 40.76% |
| No | 351 | 57.92% |
| Willingness to pay | | |
| Yes, paying collectively | 332 | 54.79% |
| Yes, paying voluntarily | 250 | 41.25% |
| No | 9 | 1.49% |

*5.2. Model 1: The Association between Willingness to Pay for Collective Cooperation Donation and Cooperation Donation Amount*

The estimated results of Model 1 in Table 3 show that the *n_eff* values are all larger than 1000, and *Rhat* values are all equal to 1, indicating Markov chain convergence. We also visualised the iterations of the simulation in the trace plots to confirm the convergence (Figure 2). Four Markov chains fluctuate around a central equilibrium after the warm-up period (after the 2000th iteration), confirming the convergence of Markov chains.

**Table 3.** Simulated posteriors of Model 1.

| Parameters | Mean | SD | *n_eff* | *Rhat* |
|---|---|---|---|---|
| *Constant* | 492.57 | 18.61 | 9351 | 1 |
| *CollectiveDonation* | 11.00 | 9.67 | 10,251 | 1 |

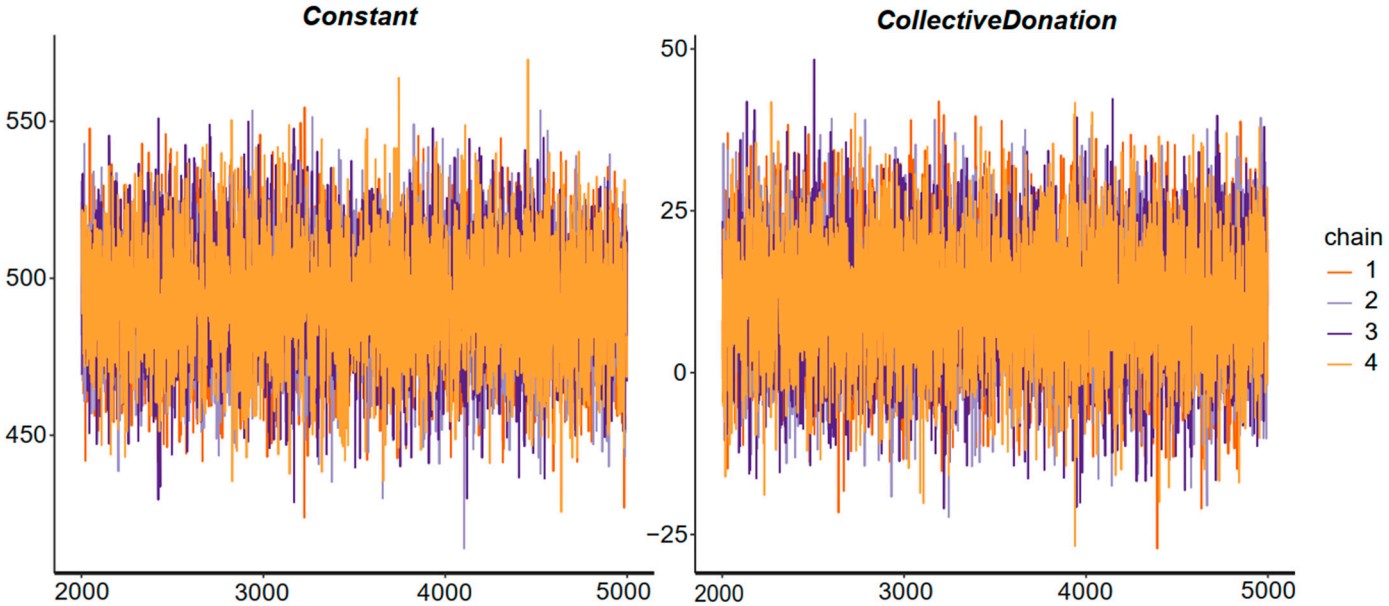

**Figure 2.** Model 1's trace plots.

The estimated results suggest that visitors thinking that cooperation donation payment should be for all climbers in principle were willing to pay more than those supporting the voluntary cooperation donation payment system ($M_{CollectiveDonation}$ = 11 and $S_{CollectiveDonation}$ = 9.67). The posterior distribution of *CollectiveDonation* coefficient is shown in Figure 3. The thick black line in the middle of the distribution illustrates the Highest Posterior Density Intervals (HPDI) or the most credible region of *CollectiveDonation*'s estimated value. Although most of the line is located on the positive side of the *x*-axis, a small portion of it is still on the opposing side. Hence, we concluded that *CollectiveDonation* has a positive impact on *DonationAmount*, although the impact is only moderately reliable.

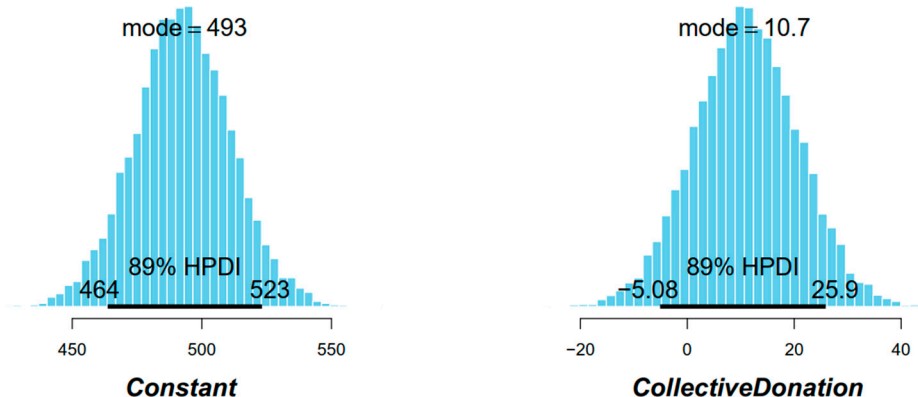

**Figure 3.** Model 1's posterior distributions.

*5.3. Model 2: Predictors of Visitors' Willingness to Pay for the Collective Cooperation Donation*

Based on the *n_eff* and *Rhat* values of Model 2 in Table 4, we can deem that all Markov chains are well-convergent. The "healthy" mixing of Markov chains in the trace plots also confirms the convergence (see Figure 4).

**Table 4.** Simulated posteriors of Model 2.

| Parameters | Mean | SD | n_eff | Rhat |
|---|---|---|---|---|
| *Constant* | −0.88 | 0.53 | 6523 | 1 |
| *Awareness* | 0.60 | 0.23 | 12,352 | 1 |
| *Sex* | −0.54 | 0.25 | 10,382 | 1 |
| *Age* | 0.03 | 0.01 | 8642 | 1 |
| *Income* | 0.19 | 0.13 | 9052 | 1 |
| *Stay* | −0.01 | 0.23 | 11,382 | 1 |
| *ClimbFrequency* | −0.17 | 0.17 | 6874 | 1 |
| *ClimbFrequency * KujuClimbExperience* | 0.01 | 0.05 | 6392 | 1 |

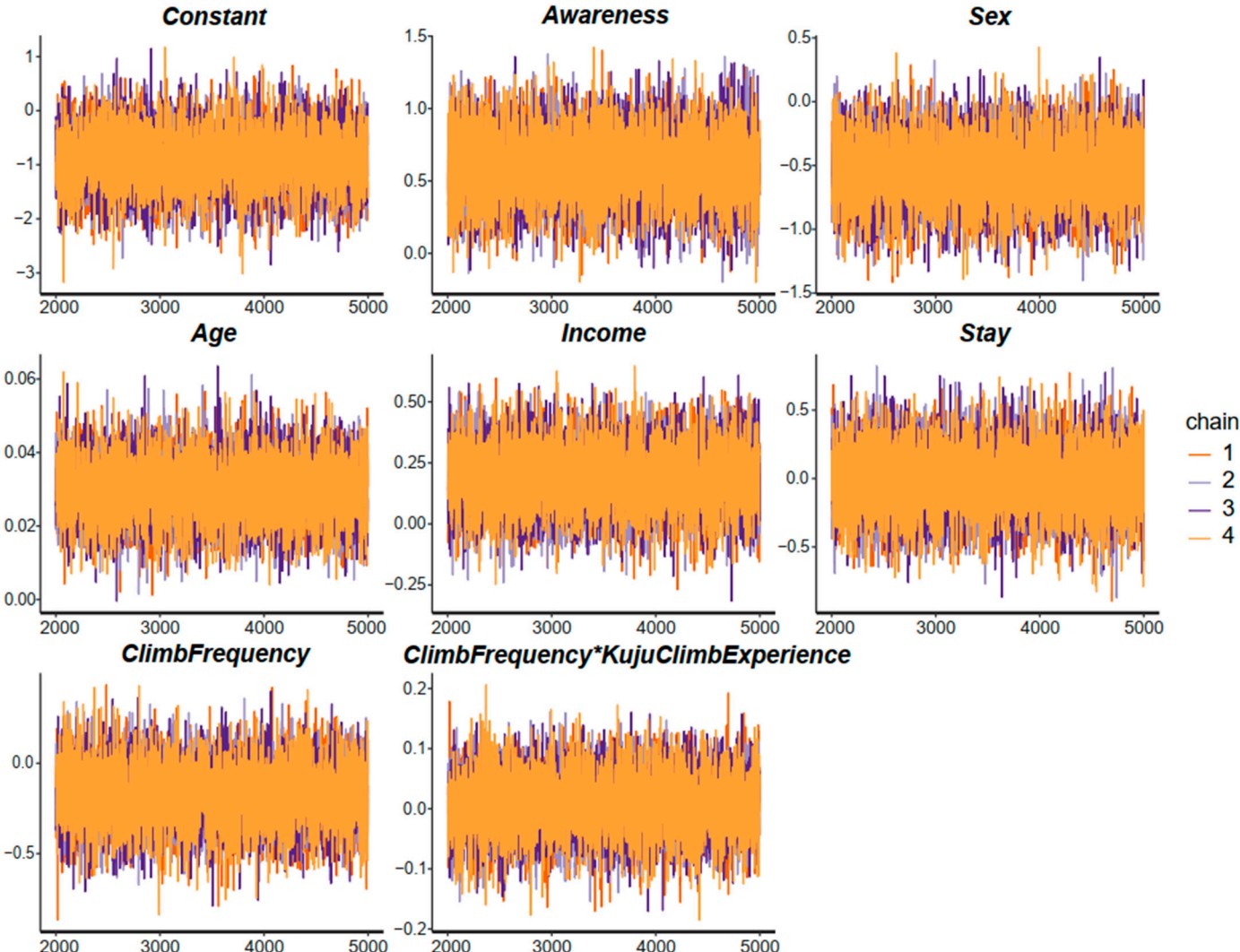

**Figure 4.** Model 2's trace plots.

We found that visitors with prior knowledge of the cooperation donation system displayed greater WTP than those without ($M_{Awareness}$ = 0.60 and $S_{Awareness}$ = 0.23). Regarding socio-demographic factors, biological sex ($M_{Sex}$ = −0.54 and $S_{Sex}$ = 0.25), age ($M_{Age}$ = 0.03 and $S_{Age}$ = 0.01), and income ($M_{Income}$ = 0.19 and $S_{Income}$ = 0.13) of the visitors could predict their WTP. In particular, females were more likely to pay the donation than their male counterparts, older visitors were more likely to pay, and higher-income visitors were more likely to pay the collective cooperation donation than lower-income counterparts. Travel style had no effect on WTP. Moreover, visitors who climbed more frequently were less willing to pay the collective cooperation donation ($M_{ClimbFrequency}$ = −0.17 and $S_{ClimbFrequency}$ = 0.17).

However, the climbing experience in Kuju did not have any particular effect on the relationship between climbing frequency and willingness to pay.

All the coefficients' posterior distributions are visualised in Figure 5. As 89% of HPDI of *Awareness*, *Sex*, *Age*, and *Income* are located entirely on either the negative or positive side of the *x*-axis, it suggests that their effects on *CollectiveDonation* are highly reliable. Meanwhile, a small portion of *ClimbFrequency*'s 89% HPDI is still on the positive side, implying that its effect on *CollectiveDonation* is moderately reliable.

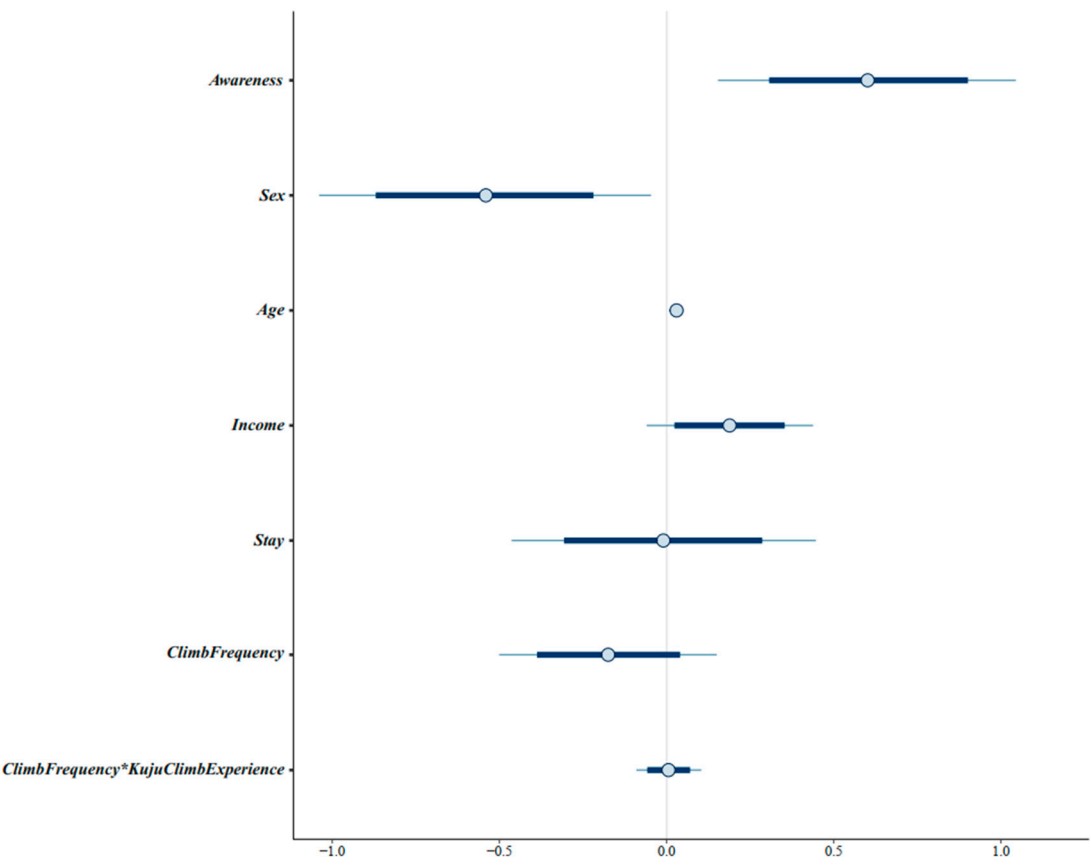

**Figure 5.** Model 2's posterior distributions. The dot in the middle of the distribution represents the mean value, while the thick blue part demonstrates the HPDI at 89% of the distribution.

## 6. Discussion

In lieu of mandatory entrance fees, donations are often collected from visitors to help finance the conservation of protected areas such as national parks, nature reserves, or wetlands. In anticipation of the expected establishment of a new conservation fund in Aso Kuju National Park, this paper investigated visitor perceptions and WTP the new donation. The results of our study suggest a relatively straightforward introduction of a JPY 500 donation as an additional revenue stream to mitigate financial shortages faced by park managers. Respondents showed little resistance to the suggested amount, with nearly all of those surveyed willing to contribute to some kind of fund and only nine respondents (1.5%) claiming they would not pay the donation.

Within the overall scope of visitor WTP, a range of socio-demographic variables could partially predict the self-stated acceptance of donations. Older, higher-income visitors appeared more likely to support the cooperation donation than their lower-income counterparts. Although a certain overlap between age and income variables could be a confounding factor, these results still contravene Diez-Gutierrez and Babri [4] finding that young and highly educated tourists were more willing to pay for an entrance fee in Norway. Above all, our results from Aso Kuju confirm the difficulty of using age as a

reliable predictor for travel expenses such as donations and entrance fees [49]. Economic benchmarks can offer higher validity, and previous studies suggest that low-income tourists are more price-sensitive [4,24,50]. In the current study at Kuju, gender was also found to be a significant predictor, with females demonstrating a higher degree of support for cooperation donation than males. This trend was in accordance with the findings from Song, Xue, Jing and Zhang [24] but is in contrast with those of Diez-Gutierrez and Babri [4].

Our results further suggest that visitors with prior knowledge of the donation were more likely to pay than new visitors who learned of the new system for the first time [42]. This echoes Reynisdottir, et al. [51] finding that respondents who had previously paid to enter alternative sites were more willing to pay a fee at a new site. Conversely, Aso Kuju visitors who were more frequent climbers in national parks manifested lower WTP, perhaps due to the perceived increased costs.

Overall, the results seem to reflect the national context since other similar donations were already introduced at Mt. Fuji, Yakushima, etc., with pilot projects akin to Kuju being trialled across other parks and protected areas [42]. Japan's Nature Park Law echoes that of Nordic countries such as Norway, where the Outdoor Recreation Act (1957) has no provision for entrance fees to protected areas [4]. For smooth implementation of the new donation, it is thus important to discuss with a range of stakeholders who need to understand the justification for the new donation policy in PAs that have traditionally been free access. In the trial phase of the program, visitors should be interviewed systematically about the amount and method of payment. The optimal amount should also be taken into consideration, along with the cost of national park maintenance.

In practice, the setup of the new donation system could entail considerable costs to establish and operate a collection system vis-a-vis extra human resources and gates or checkpoints. At Mt. Fuji, where a similar system has been in operation for over a decade, a considerable proportion of collected funds are still spent on staff wages (personal correspondence). In Kuju's case, human resource costs and undue site hardening, such as the construction of additional infrastructure, could be mitigated by the voluntary nature of the donations programme. However, price sensitivity and equity could likely become an issue due to the large area and crisscrossing access routes across the Handa Plateau.

## 7. Conclusions

Protected areas are pioneering new financial instruments, such as visitor donations, to supplement conservation budgets and enhance effective management. Japan's national parks do not routinely require visitors to pay an "entrance ticket" or pre-determined fee. However, certain destinations such as Yakushima and Mount Fuji have introduced a discretionary donation system to support environmental conservation and climber safety. At Fuji, the suggested amount of JPY 1000 is collected at the trailhead but can also be paid directly online or at convenience stores across Japan [42]. This paper investigated visitor perceptions of a parallel JPY 500 conservation donation system under consideration in Aso Kuju National Park, Japan. A trailhead survey in autumn 2022 gauged visitors' WTP at the expected JPY 500. Based on these findings, the collection of the new donation was largely acceptable to visitors and could help to safeguard quality nature-based tourism opportunities. The new donations could supplement existing budget allocations for park management, offering an ongoing "crowdfunded" alternative to periodic, "pop-up" campaigns that focus on particular maintenance and management flashpoints such as trails, traffic controls, or toilets [51].

Variation between domestic and international visitors' core values and WTP poses new challenges for site managers, making micro-level monitoring of consumer behaviour indispensable. This research draws on primary data collected from visitors in an intercept survey conducted in autumn 2022 to address this practical and academic gap in the literature.

## 8. Limitations

As a single case study site, our survey focussed on the two most popular Kuju trailheads in the autumn season. Before a donation policy is implemented at Aso Kuju, different areas, seasons, and types of attractions across the national park should be investigated to provide representative results across diverse visitor segments [32]. Moreover, the operation costs of collecting and processing the donation should also be considered as part of a holistic cost-benefit analysis to determine the price level [51].

The self-stated WTP for Aso Kuju National Park revealed personal economic valuations of the public good but could also reflect differences in predictor variables such as age or income. In reality, WTP can vary according to visitors' connectedness with the protected areas (e.g., iconic places or landscapes) and their underpinning values, goals, and worldviews about nature [52,53]. However, these aspects were beyond the scope of the current study. Thus, future studies can focus on studying how visitors' human–nature nexus can affect their WTP. Insights from this direction can support policymakers' and practitioners' efforts to build an eco-surplus culture among urban residents, potentially improving their likelihood to contribute to financing protected areas [54].

**Author Contributions:** Conceptualisation, T.E.J.; methodology, M.-H.N.; software, M.-H.N.; validation, T.E.J.; formal analysis, M.-H.N. and T.E.J.; investigation, T.E.J. and D.X.; resources, T.E.J. and T.K.; data curation, M.-H.N. and D.X.; writing—original draft preparation, T.E.J.; writing—review and editing, T.E.J. and M.-H.N.; visualisation, M.-H.N. and T.E.J.; supervision, T.E.J.; project administration, T.K.; funding acquisition, T.E.J. All authors have read and agreed to the published version of the manuscript.

**Funding:** This research was funded by the Japanese Ministry of Environment..

**Data Availability Statement:** Restrictions apply to the availability of these data. Data were obtained from the Japanese Ministry of Environment and are available from the authors with the permission of the Japanese Ministry of Environment.

**Acknowledgments:** We gratefully acknowledge technical support by Kelvianto Shenyoputro.

**Conflicts of Interest:** The authors declare no conflicts of interest.

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
