# Peer review of "Visitors’ Willingness to Pay for Protected Areas: A New Conservation Donation in Aso Kuju National Park"

_conservation, doi:10.3390/conservation4020014_

Round 1
Reviewer 1 Report
Comments and Suggestions for Authors
This paper investigates visitor perceptions of one such system, a new conservation methodology funded through donations, under consideration in Aso Kuju National Park - South west Japan. Towards this end, an on-site survey at two trailheads in Autumn 2022 gauged visitors’ willingness to pay (WTP) the expected 500 JPY donation was cased. The analysis of the suggested model used Bayesian linear regression to look for significant predictors of WTP.
Interesting mathematical considerations and transformations were used to support the handling of the research questions and related hypotheses of the paper.
General Comment
The structure of the paper is quite reasonable and well balanced.
Specific Comments
Comment 1
Line 223: It isn’t clear the reason why those 8 variables were chosen for model 2
Comment 2
Fig.3 : The presentation of the graphs-trace is not easily perceived
Fig.5: The presentation of the graphs-trace is not easily perceived
Comment 3
Literature Review and especially 2.2. “WTP for protected areas: the global context” should be enriched

Author Response
Thank you very much for your constructive feedback! Please see our detailed responses to your comments in the attached file.

Reviewer 2 Report
Comments and Suggestions for Authors
Overall I found the manuscript clear and well presented. I'd recommend including a comparison of the demographics presented in this study to comparative census data (Japan national level demographic profile, or possibly visitor stats from the park if available), just to give some context for the reader of how well your sample matches with population under consideration.
Author Response

(The authors gave the same response as above.)

Reviewer 3 Report
Comments and Suggestions for Authors
Financing protected area management is an important topic considering the global underfunding of protected areas. This paper looks at the willingness to pay for visitation to a national park in Japan and presents results based on the socio-economic characteristics of the respondents.
It wasn’t clear in the paper whether this donation system was proposed or actually in place. The title implies it is yet the paper is about willingness to pay, thus implies it is exploratory. This needs to be clarified.
There is reference to “pay the donation collectively rather than voluntarily” in the abstract and in the results yet these terms are not defined anywhere nor is it mentioned in the discussion. This is a major issue that needs to be addressed. Specific additional comments below:
Line: 19: “pay the donation collectively rather than voluntarily” – unclear what this means
21: Not sure what ‘collectively’ means in the following context: “pay the donation collectively”
34: Why just economic instruments? Why not regulatory instruments?
79: does the $ value here represent USD? If so make that explicit
82-83: what does ‘maximise’ mean in this context? Wouldn’t a Rs40 or 100 per person fee also ‘maximise’ park revenue?
91-92: “These studies collectively underscore the global 91 significance of WTP analysis as a research domain” – how? All it says is that a number of studies have been undertaken, nothing of the global significance
127-135: a mix of common names and scientific names are being used here. Suggest use both a common name and scientific name for each species.
155: suggest change ‘repeaters’ to ‘repeat visitors”
179-180: “two professors, together with five trained data collectors.” – the authors of this paper? Need to define.
210: ‘beautiful’ not needed here
219: “collectively or voluntarily”. These terms are used later in the results but have not been defined. They need to be earlier in the manuscript.
261: replace “from 40’s to 60’s” with “from 40 to 69”
262: “Tourists with more than” should be “Tourists earning more than”
Figure 2 and Table 2: Age. Better to have the exact age ranges, e.g. 40-49, 50-59 rather than 40’s and 50’s
Figure 2: Isn’t all this information in Figure 2 also in Table 2? If so you don’t need both.
317-318: “The dot in the middle of the distribution represents the mean value, while the thick blue part demonstrates the HPDI at 89% of the distribution.” – this information needs to go in the figure caption, not the text,
328: are wetlands as protected area by default in Japan?
348: “contravention” isn’t the right term – perhaps “is in contrast with” is better?
350: “newbies”. This terms has no place in a scholarly journal.
355: “the results seem to reflect the national context “. This is unclear to me.
Comments on the Quality of English Language
Mostly fine. Suggested changes above
Author Response

(The authors gave the same response as above.)

Round 2
Reviewer 3 Report
Comments and Suggestions for Authors
Thank you for addressing all of my comments. Perhaps the only thing I was surprised with was the decision to delete Table 2 and retain Figure 2 based on my comments. I thought there was actually more information in the Table than the Figure so thought the other option (delete Figure 2 and retain Table 2) might have been more appropriate. However, as long as all of the results from Table 2 are already in the text then this should be okay,
Author Response
Thank you very much for your constructive feedback! Please see our response in the attached file.
